# CT-Based Radiomics to Predict *KRAS* Mutation in CRC Patients Using a Machine Learning Algorithm: A Retrospective Study

**DOI:** 10.3390/biomedicines11082144

**Published:** 2023-07-29

**Authors:** Jacobo Porto-Álvarez, Eva Cernadas, Rebeca Aldaz Martínez, Manuel Fernández-Delgado, Emilio Huelga Zapico, Víctor González-Castro, Sandra Baleato-González, Roberto García-Figueiras, J Ramon Antúnez-López, Miguel Souto-Bayarri

**Affiliations:** 1Department of Radiology, Complexo Hospitalario Universitario de Santiago de Compostela, 15706 Santiago de Compostela, Spain; jacobo.portp.alvarez2@sergas.es (J.P.-Á.); rebeca.aldaz@rai.usc.es (R.A.M.); emilio.huelga.zapico@sergas.es (E.H.Z.); roberto.garcia.figueiras@sergas.es (R.G.-F.); miguel.souto@usc.es (M.S.-B.); 2Centro Singular de Investigación en Tecnoloxías Intelixentes da USC (CiTIUS), Universidade de Santiago de Compostela, 15705 Santiago de Compostela, Spain; manuel.fernandez.delgado@usc.es; 3Department of Electrical, Systems and Automation Engineering, Universidad de León, 24071 León, Spain; victor.gonzalez@unileon.es; 4Department of Pathology, Complexo Hospitalario Universitario de Santiago de Compostela, 15706 Santiago de Compostela, Spain; jose.ramon.antunez.lopez@sergas.es

**Keywords:** *KRAS* mutation, colorectal cancer, texture analysis, radiomics, radiogenomics

## Abstract

Colorectal cancer (CRC) is one of the most common types of cancer worldwide. The *KRAS* mutation is present in 30–50% of CRC patients. This mutation confers resistance to treatment with anti-EGFR therapy. This article aims at proving that computer tomography (CT)-based radiomics can predict the *KRAS* mutation in CRC patients. The piece is a retrospective study with 56 CRC patients from the Hospital of Santiago de Compostela, Spain. All patients had a confirmatory pathological analysis of the *KRAS* status. Radiomics features were obtained using an abdominal contrast enhancement CT (CECT) before applying any treatments. We used several classifiers, including AdaBoost, neural network, decision tree, support vector machine, and random forest, to predict the presence or absence of *KRAS* mutation. The most reliable prediction was achieved using the AdaBoost ensemble on clinical patient data, with a kappa and accuracy of 53.7% and 76.8%, respectively. The sensitivity and specificity were 73.3% and 80.8%. Using texture descriptors, the best accuracy and kappa were 73.2% and 46%, respectively, with sensitivity and specificity of 76.7% and 69.2%, also showing a correlation between texture patterns on CT images and *KRAS* mutation. Radiomics could help manage CRC patients, and in the future, it could have a crucial role in diagnosing CRC patients ahead of invasive methods.

## 1. Introduction

The carcinogenesis of colorectal cancer is a heterogeneous process encompassing a series of genetic, epigenetic, and molecular changes in the cells that line the colonic mucosa [1,2]. These changes are influenced by dietary, environmental, and microbiotic factors and the host’s immune response [3,4]. The successive activation of oncogenes (*KRAS*, *NRAS*, *BRAF*, *PIK3CA*, *ERRB2*) while inactivating tumor suppressor genes (*APC*, *TP53*, *PTEN*, *TGF-β*, *DCC*) guides the adenoma–carcinoma transition [5,6]. *KRAS* is a gene of the *RAS/MAPK* pathway. RAS proteins are a family of proteins expressed in all cells within the intracellular cascade associated with tyrosine-kinase receptors. This pathway stimulates cell proliferation, differentiation, adhesion, apoptosis, and migration [7]. Up to 60–80% of colorectal cancers overexpress EGFR (epidermal grow factor receptors), which are tyrosine-kinase receptors, and this is an important component in the initiation and progression of colorectal cancer [8,9]. Anti-EGFR antibodies (cetuximab or panitumumab) have a therapeutic effect in patients with colorectal cancer. When there is a mutation in this pathway, such as the *KRAS* mutation, these therapies cannot be employed because they confer resistance to EGFR antibodies. The *KRAS* mutation is present in 30–50% of colorectal cancers [9], and this mutation is associated with worse survival, so it is considered a negative prognostic factor [10,11].

Radiomics is the transformation of radiological images into structured data that can be used to support decision-making in day-to-day clinical practice. Data that are not visible to the human eye are taken into account through radiomic analysis of a radiological image [12,13]. The development of radiomics is more pronounced in the field of oncology. There have been several studies published in recent years using radiomics in different cancers and using different imaging modalities, such as magnetic resonance imaging (MRI), ultrasound, and CT.

This article aims to predict the presence of *KRAS* mutation in colorectal cancer patients using a CT-based radiomics model.

A second objective of this article is to explore the performance of clinical data and whether it can improve the radiomic model when both are combined.

To achieve these objectives, it is necessary to extract the radiomics features from the CT images and automatically classify the patients as *KRAS*+ or *KRAS*− using machine learning algorithms belonging to different classifiers families, such as support vector machine, neural network, linear discriminant analysis, decision trees, and ensembles, among others. Data such as tumor location, presence of hepatic or pulmonary metastases, as well as tumor stage and differentiation are included in the analysis. The results were compared with the anatomopathological analysis of the tumor using Cohen’s Kappa statistic, sensitivity, and specificity.

Anatomopathological analysis of the tumor is the gold standard for determining *KRAS* mutation, but it is an invasive test, and it only analyses a portion of the tumor. Radiomics is a non-invasive method that can help determine *KRAS* status by localizing the area of the tumor most likely to have a *KRAS* mutation and guiding the biopsy.

## 2. Materials and Methods

### 2.1. Radiomics Workflow

A radiomics workflow consists of five sequential steps: image acquisition, pre-processing, region of interest segmentation, feature extraction, and analysis (Figure 1) [13,14].

### 2.2. Patient Selection and Obtaining Imaging

For this retrospective study, 56 patients from the Santiago de Compostela Health District were selected. The inclusion criteria were defined as follows: (1) colorectal cancer patients with anatomopathological confirmation of *KRAS* status by biopsy between 2016 and 2019 (30 *KRAS*+ and 26 *KRAS*-patients, respectively); (2) intravenous CECT performed before any treatment; (3) CT images with a slice thickness of less than 5 mm. The exclusion criteria were as follows: (1) patients with colorectal cancer in which the anatomopathological analysis was performed after any type of treatment; (2) CT with a slice thickness other than that specified; (3) patients with a tumor difficult to delineate. The conduct of this research was approved by the Ethics Committee.

### 2.3. Segmentation

Manual segmentation was performed by an expert abdominal radiologist. The software used for the segmentation was the Sectra IDS7 visualization program (version 24.2.14.6022, Linköping, Sweden), used routinary by the radiology department of the Clinic Hospital of Santiago de Compostela. 

Three slices of the tumor were selected: the slice with the largest tumor area (central slice) and the slices immediately cranial and caudal to that central slice. For each slice, 4 images were obtained, 2 of them with the tumor manually segmented. A total of 12 images were obtained for each patient. The images were saved in “.tiff” format (Figure 2 and Figure 3).

### 2.4. Pre-Processing and Feature Extraction

Texture is a visual image property related to the spatial distribution of the gray level of pixels [15], which may be used for image classification. Feature extraction algorithms transform an image or region of interest (ROI) in the image into a feature vector, which will be used to carry out the classification. Some of the techniques provided in the literature can only be applied to rectangular or even squared regions, and they are not suitable for our problem, in which the cancer tumors are irregular regions. In previous works, some popular texture extraction techniques were adapted to operate over irregular ROIs [16]. For example, frequency techniques, such as Gabor or Fourier filters, are global techniques and cannot be adapted to operate on irregular regions. Among the statistical techniques we used in this study are Haralick coefficients and local binary patterns (LBP), which are described in our previous work [17]. Let *G* = {0, 1, …, *N_g_*−1} be the number of grey levels, *S* a finite set of pixels specifying the region of interest (ROI) to be analyzed (in our case, the tumor), and *I*(*x*, *y*) ∈ *G* the grey level in the pixel (*x*, *y*) ∈ *S*. To compute Haralick coefficients [17], the gray level cooccurrence matrix, M, counts the occurrence of pixel (*x_i_*, *y_i_*) ∈ S and (*x_j_*, *x_j_*) ∈ *S* with a gray level *g_i_* and *g_j_* ∈ *G* in a specific orientation *θ* and scale, i.e., different distance, *d* = {1, 2, 3}, from one pixel to each other. The matrices M, of dimension *N_g_* × *N_g_*, are calculated for different orientations *θ* and scales *d*. Normally, the matrices M for different orientations are averaged in order to achieve rotation invariance. The energy, correlation, contrast, homogeneity, and entropy are derived from the matrix M for each scale. Eight Haralick vectors Hf*_dm_* are calculated varying the distance *d*, to be *d* = {1, 2, 3} pixels, alongside with another vector concatenating the previous three distances, labeled as *d* = 123, and the number of Haralick coefficients *m* = {4, 5} depending of entropy coefficient is included or not: {Hf*_dm_*, *d* = 1, 2, 3, 123, *m* = 4, 5}.

The local binary patterns (LBP) algorithm extracts the dependence of pixels in a neighborhood by comparing the gray level of the central pixel with the surrounding ones [18]. Among the variants proposed in the literature, we used the LBP uniform patterns, i.e., patterns which a limited number of transitions from 0 to 1 or vice versa, lower, or equal to two. Four texture feature vectors, called LBP*_R_*, one for each radius *R* = {1, 2, 3}, were used with a neighborhood of *p* = 8 pixels. The LBP*_R_* contains 10 features for each scale or radius, representing a histogram of the number of transitions in the pixels (*x*, *y*) ∈ *S.* The feature vector LBP_123_, with 30 features, is the concatenation of the previous three LBP*_R_* vectors: {LBP*_R_*, *R* = 1, 2, 3, 123}.

The discrete wavelet transform (DWT) is another very popular spectral technique for texture extraction, which is normally applied on a squared region whose side is a power of 2. Multi-scale decomposition is obtained by applying recursively low-pass and high-pass filters and downsampling to the image. Statistical measures over the transform coefficients for each sub-band and decomposition level are normally used to encode texture information. The measurements used are mean, energy, entropy, and standard deviation calculated in the pixels (*x*, *y*) ∈ *S*. We computed the vector DWT using three levels of decomposition and calculated the energy, entropy, mean, and standard deviation measures over all subbands, developing a vector with 52 features.

We also derived texture descriptors from the wavelet decomposition in different ways (see our reference [19] for a detailed description): Applying Haralick coefficients over the different wavelet decomposition levels using a distance *d* = 1, due to the scale being implicitly included in the downsampling. We computed four vectors, called WDCF*_fm_,* varying the number of Haralick coefficients *m* = {4, 5} and the type of decomposition *f* = {LL, All} using a distance *d* = 1: {WDCF*_fm_*, *f* = LL, All, *m* = 4, 5}.Another way to capture multiscale information would be calculating the cooccurrence matrices on the first level of decomposition. We computed the following six vectors (WCF*_dm_*) varying the distance *d* = {1, 2, 3} to calculate the cooccurrences matrices and the number of Haralick’s coefficients *m* = {4, 5}: {WCF*_dm_*, *d* = 1, 2, 3, *m* = 4, 5}.The multiscalar information can also be captured by calculating the LBP signature using a radius of one pixel over low-pass wavelet decompositions of the original image. Specifically, the four vectors LBP*_s_* considering one level of wavelet decomposition *s* = {1, 2, 3} were computed, developing feature vectors of 10 features. The concatenation of the three previous vectors is called LBP_123_ with 30 features: {LBP*_s_*, *s* = 1, 2, 3, 123}.

### 2.5. Machine Learning Models

Machine learning is widely used in medicine to predict different indicators. Our attempt is the prediction of *KRAS* status, in which there are two possible labels, patients with *KRAS*+ or *KRAS*−, the former considered as a “positive event” to be detected. So, this prediction is a case of binary classification (i.e., with 2 classes). We selected a reduced collection of best-performing classifiers of different families proposed in the literature. These classifiers are trained to learn from the input data (texture features and patient clinical information, see below) how to predict the output (*KRAS*+ or *KRAS*−). In this training process, the classifier used a collection of examples composed of input data and the desired output (gold standard). The trained classifier was expected to predict, with more or less reliability, the genetic disease of unseen patients. In the current experimental work, we used a collection of 34 classifiers implemented in the programming languages MATLAB, Octave, Python, and R, belonging to the families: support vector machine, neural network, decision tree, bagging, ensemble, and linear discriminant analysis, among others. Table 1 lists the information about the classifiers used in this work.

The classifier performance was assessed by Cohen’s kappa value, which measures the agreement between the true and predicted class labels excluding the agreement by chance [20]. Other performance metrics are the accuracy, sensitivity, or recall, specificity, positive predictivity, or precision, F1, and area under the receiver operating curve (AUC). See our reference [17] for a description of these measures.

## 3. Experimental Setup and Results

### 3.1. Experimental Setup

The information related to each patient is of two types: Vector clinical, containing information related to the patient’s life and their histopathology status before any treatment, composed of the following nine values: liver metastasis, pulmonary metastasis, sex (dicotomical variable), age, location of the tumor, T staging (0, 1, 2, 3, or 4), N staging (0, 1, or 2), M staging (0 or 1), and tumor differentiation (stages 1 to 4).Texture feature vectors, with features extracted from each slice of the CT as explained in Section 2.4; in our case, three slices of the tumor for each patient (56 patients multiplied by three cuts per patient).

Classifier performance was assessed using the leave-one-patient-out cross-validation approach. This methodology uses one patient (i.e., three cuts) to test the model and the remaining ones to train the model and to adjust its tunable hyper-parameters (see in Table 1 the values of the tunable parameters of each model). All the inputs are pre-processed to have zero mean and a standard deviation of one. This process is repeated as many times as there are patients, each time using a different test patient. Finally, the performance is calculated by comparing the label predicted by the classifier and the gold standard for determining *KRAS* mutation for each patient. In the case of texture feature vectors, we obtained an input vector for each cut of the patient’s tumor and then a classifier output (or prediction) for each cut. Classifier prediction was selected as the most voted among the three predictions, one for each cut.

### 3.2. Results

We developed experiments applying the 34 classifiers as input: (1) the clinical vector; (2) the 27 texture feature vectors; (3) combinations of the clinical vector and the 27 texture feature vectors. The texture feature vectors used are: eight Haralick vectors Hf*_dm_*, {Hf*_dm_*, *d* = 1, 2, 3, 123, *m* = 4, 5}, four LBP*_R_*, {LBP*_R_*, *R* = 1, 2, 3, 123}, four LBP*_s_*, {LBP*_s_*, *s* = 1, 2, 3, 123}, vector DWT, six WCF*_dm_*, {WCF*_dm_*, *d* = 1, 2, 3, *m* = 4, 5}, and four WDCF*_fm_* vectors {WDCF*_fm_*, *f* = LL, All, *m* = 4, 5}. Overall, we performed 34·(1 + 27 + 27) = 1870 experiments. 

Table 2 shows a list of the top 10 best combinations of feature vectors and classifiers to predict *KRAS* mutation. The highest kappa value (53.7%) and accuracy (76.8%) were achieved by the AdaBoost (adaptive boosting ensemble) classifier implemented in Python, using the clinical vector. The best performance using only feature texture vectors (image information) was provided using the combination of wavelet and Haralick’s coefficients (feature vector WDCF*_fm_* with *f* = All bands and *m* = 4), achieving kappa = 46% and accuracy = 73.2%. The combination of clinical and imaging information did not exceed the results achieved by the clinical information alone. 

A compilation of the complete results (minimum, median, mean, and maximum kappa achieved by each classifier over all the datasets) is reported in Appendix A. The highest median, mean, and maximum kappa values were achieved, respectively, by ctree (median of 27.6%), lasso (mean of 15%), and AdaBoost (maximum of 53.7%). Appendix A of the Appendix A plots the kappa distribution (as a box plot) for each classifier over the datasets, sorted by decreasing median values. ctree and the four implementations (in MATLAB, Octave, Python, and R) of lda achieved the first positions.

Table 3 shows the confusion matrix for the best performance using clinical and imaging information (rows identify true class labels, named Biopsy, while columns identify predicted labels, named Computer). Although the differences between both confusion matrices are small, they lead to a large difference in kappa, 53.7% and 46% using *clinical* (up) and WDCF*_fm_* (down), although the difference in accuracy (76.8% and 73.2%) is smaller. In the best result, achieved by AdaBoost using the clinical vector (up), the terms outside the diagonal (five and eight false negatives and positives, respectively) are much smaller than terms in the diagonal (22 and 21 true positives and negatives, respectively). 

Other performance metrics for the best classification are reported in Table 4. Specifically, AdaBoost achieved high sensitivity (73.3%), specificity (80.8%), and area under the receiver operating curve (ROC); that is, (in %) 77.8%. The sensitivity and specificity of the best model (kappa = 53.7%) were calculated from the upper confusion matrix in Table 3. The left panel of Figure 4 plots this curve, which is quite near the upper left corner that identifies the ideal classification. The red square locates the working point achieved by AdaBoost. The right panel plots the lift curve, where the black line inside the gray shadowed area is also fairly near the left border of this area, identifying the ideal classification.

In order to analyze the behavior of different texture feature families, Table 5 shows the best performance achieved by a vector of each descriptor family. The highest kappa (46%) and accuracy (73.2%) were achieved by the combination of Haralick coefficients and wavelets using all bands (WDCF*_fm_* vector with *f* = All and *m* = 4 or m = 5) using the rpart classifier (recursive partitioning decision tree), implemented in the R language. Indeed, four of the six best results in Table 5 were achieved by classification trees (ctree and rpart). The other way to compute Haralick coefficients over wavelet decomposition (vectors WCF*_dm_*) achieved much lower performance (kappa = 28.2%). The local binary patterns (LBP*_R_* and LBP*_S_* vectors) provided similar results (kappa = 35% and 36.9%, respectively) but also much lower than the WDCF*_fm_* vector. The worst texture descriptor was the DWT vector (kappa = 12% using diagonal linear discriminant analysis). 

Table 6 shows the performance achieved using clinical and texture feature vectors concatenated as input to the classifier. The performance increased in almost all the texture families, but it was still lower than the performance achieved using only the clinical vector. The highest kappa was achieved by the combination of clinical and Hf*_dm_* vectors (using *d* = 123 and *m* = 4) and the mlp (multilayer perceptron) classifier implemented in Python. Nevertheless, all the combinations provided quite similar results (kappa value higher than 42%), except for DWT (kappa = 34.7%) and WDCF*_fm_* (38.8%). It is a surprise that WDCF*_fm_* concatenated with clinical vector decreased its performance compared to WDCF*_fm_* alone (kappa = 46% in Table 5 and 38.8% in Table 6).

## 4. Discussion

This study demonstrates that it is possible to predict *KRAS* mutation in CRC patients using CT-based radiomics features. There is a relationship between the quantitative features obtained from the images and the *KRAS* oncogene mutation. The value of Cohen’s kappa coefficient shows that the relationship is not simply justified by chance. Compared to our previous investigations, we increased the number of patients and tried a larger and more diverse collection of classifiers, including patient clinical features. In the current work, the results achieved using only clinical information, only radiomics information, and combined (clinical and radiomics) were similar but slightly better using only clinical information. This demonstrates that clinical variables also provide useful information that can be unified with radiomic data to create a more effective combined model. The implementation of clinical data into radiomic studies has been a growing field in recent years. Studies that combine clinical and radiomics features are becoming increasingly common. These studies often show improved results with clinical—radiomic association. For instance, Yuntai Cao et al. [21] developed a model that combines radiomic parameters with other clinical parameters such as age, CEA (carcinoembryonic antigen) level, and clinical stage for the prediction of *KRAS* status. They achieved results similar to ours. Their area under the curve (AUC), sensitivity, and specificity were higher with the combined model (77.2%, 79.2%, and 64.6%, respectively). Our results, together with those published in the literature, reflect the need for further research into the development of a combined model and the search for the best clinical–radiomic combination.

Studying *KRAS* status is an expanding field. There were numerous studies published on this topic between 2018 and 2022. In December 2022, Jia et al. [22] performed the first meta-analysis on this topic, including 29 articles published between February 2014 and March 2022. Approximately 60% were recent (2020 and 2021). This review reflects the multiple imaging modalities to which *KRAS* status analysis is applicable (CT, PET, and MRI). MRI was the most widely used, but the meta-analysis concluded that the diagnostic performance of CT is higher. Only one prospective design was included. The main criticisms of meta-analysis studies can be summarized in two aspects: low quality and heterogeneity. The sample size and segmentation methods are sources of variability between studies. The conclusion is that radiomics is at an early stage in terms of predicting *KRAS* status; therefore, prospective multicenter studies with standardized protocols are needed to achieve effective implementation in routine clinical practice.

Comparing with other similar studies, several aspects of our research should be highlighted. All the studies compared are retrospective. The sample size of our study is similar to other similar published studies, such as that of Taguchi et al. [23], who created a model to predict *KRAS* mutation in CRC. They obtained an AUC between 40% and 70% with 40 patients. Slice thickness varies between 1 and 5 mm, depending on the study. *KRAS* is the oncogene analyzed in all the studies, but Yang et al. [24] analyzed *NRAS* and *BRAF* mutations as well. They obtained an AUC, sensitivity, and specificity of 86.9%, 75.7%, and 83.3%, respectively. Yu Li et al. also sought to detect perineural invasion. They obtained an AUC of 79.3% and 86.2% in the prediction of perineural invasion and *KRAS* mutation [25].

Anatomopathological analysis will remain the gold standard for mutational analysis. However, it has limitations that could be solved if complemented by radiomic analysis. Radiomics based on CT images would cover the entire tumor area and possible metastatic sites, thus avoiding the false negatives associated with analyzing a single tumor fragment [26]. There is also a percentage of patients with primary resistance to anti-EGFR monoclonal antibodies. In addition, almost all of those who initially respond to these therapies will eventually become refractory to treatment [27]. Therefore, serial radiomic CT scans throughout the course of the disease would offer the possibility of detecting new mutations in *KRAS* that cause this resistance.

The limitations of our study do not differ from those mentioned by other authors. Firstly, we started from a small sample size obtained retrospectively from the records of a single center. Secondly, despite having an advanced electronic medical record system, it is difficult to filter which patients could meet the inclusion criteria for the study, as there are no databases of patients diagnosed with colorectal cancer in our autonomous community. This could lead to patient selection bias and hinder the applicability of radiomics in the future. Regarding technical parameters, over the years, the reconstruction of the images has evolved. The first slices compiled for this study were 5 mm and became less than 2.5 mm. The trend is to use increasingly thinner slices, as this is one of the parameters that provide the greatest variability in radiomic studies. Finally, segmentation is manual, which is associated with a high time consumption in the delimitation of the regions of interest and a high level of subjectivity.

Lastly, we should consider extending the mutational analysis to other genes such as BRAF, NRAS, mismatch repair genes, and even to other sites such as metastases. This approach would be more integrative and would facilitate the use of radiomics throughout the entire period of the disease, from diagnosis to the evolution of metastases.

## 5. Conclusions

In this manuscript, we were able to predict the *KRAS* oncogene mutation using three different prediction models with TC-based radiomics features and clinical data. 

The first prediction model considers only radiomic data, and its results have improved over those obtained in our previous work, with a Cohen’s kappa and accuracy of 46% and 73.2%, respectively. These results demonstrate the correlation between texture patterns and *KRAS* mutation status. 

The second prediction model is a combined model that incorporates clinical and radiomic data from patients. The results obtained with this model were similar to those of the radiomic model, with improvement in some classifiers, but with the same results in terms of best accuracy and best Cohen’s kappa.

Finally, the third prediction model was based on clinical data alone. The clinical parameters obtained were the presence of liver or lung metastases, tumor location, N and M staging, and tumor differentiation. The best results achieved by the AdaBoost ensemble classifier were an accuracy and Cohen’s kappa of 53.7% and 76.8%, with a sensitivity and specificity of 73.3% and 80.8% and AUC of 77.8%. 

The results obtained with the different models showed that: (1) there is a correlation between radiomic features and *KRAS* mutation status, with the CT-based radiomic model predicting *KRAS* oncogene mutation with 73% accuracy; (2) the clinical data provide important information that should be considered in our trials. The results obtained, together with those of other research groups, show that they can improve the performance of the model based on radiomic data alone. Of course, the small size of the sample (56 patients) is a limitation of this study.

*KRAS* mutation detection is based on anatomopathological analysis and does not take into account the patient’s clinical data, but the results show that radiomics features and clinical data can provide valuable information for the management of patients with CRC. The clinical data should also be incorporated in the form of a combined model, as reflected in the literature consulted, where combined models perform best. In this case, the reason why our results are better with the clinical model than with the combined model requires further research.

The *KRAS* oncogene is a prognostic factor in patients with colorectal cancer. Anatomopathological analysis of these tumors is performed after the tumor has been diagnosed and localized by a surgical procedure (excision or biopsy). In addition, anatomopathological analysis only covers part of the entire lesion, which may not be the area where the mutation is present. Radiomic analysis offers two possible solutions to these drawbacks of classical pathological analysis. First, it is a non-invasive analysis that can be performed at the time of diagnosis. Second, radiomic analysis can cover the entire tumor tissue and localize the areas of the tumor that are most likely to carry the mutation. At this stage, radiomics are not intended to replace pathological analysis but to guide it to the areas of the tumor where a biopsy would be most cost-effective.

On the other hand, radiomic analysis has a number of limitations. Firstly, there is still a need to standardize the feature acquisition process and the automatic classification methods used. For this, it is essential to share the methodology used and to search for the best-performing models. Another limitation of radiomic analysis is the lack of prospective studies. In this sense, the application of radiomics in tumor pathology has not contributed to the possibility of carrying out these prospective studies because the diagnostic pathways in oncology patients are very well optimized. Therefore, the use of radiomics in pathologies where diagnostic pathways are not well defined or standardized could facilitate the possibility of prospective and standardized studies. Finally, although manual segmentation is the gold standard, it is very time-consuming for the radiologist. Therefore, semi-automated or automated segmentation should take over and become the reference in future studies.

## Figures and Tables

**Figure 1 biomedicines-11-02144-f001:**
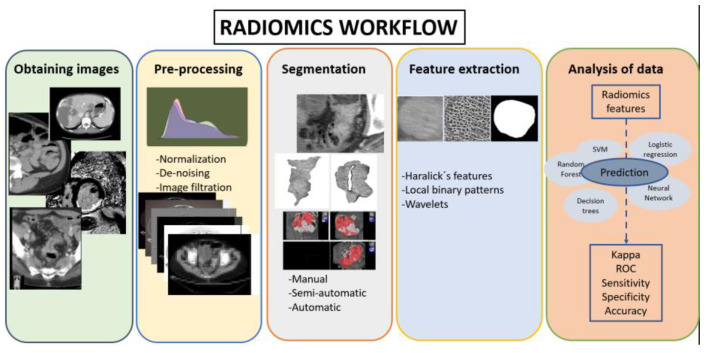
Radiomics workflow.

**Figure 2 biomedicines-11-02144-f002:**
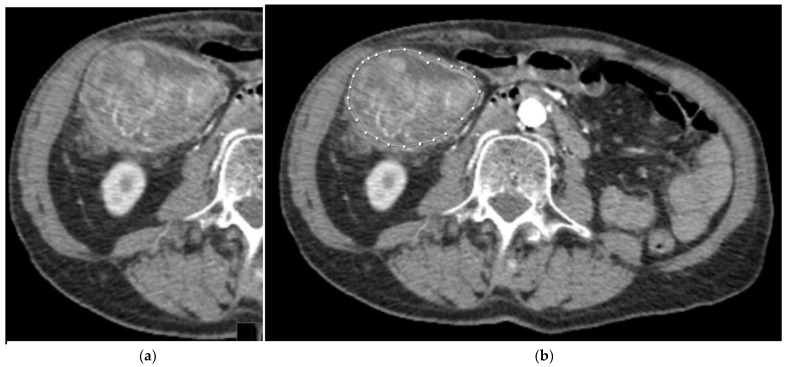
A 58-year-old patient with *KRAS*-mutated CRC. (**a**) Tumor without any segmentation. (**b**) Tumor manually segmented by an expert abdominal radiologist.

**Figure 3 biomedicines-11-02144-f003:**
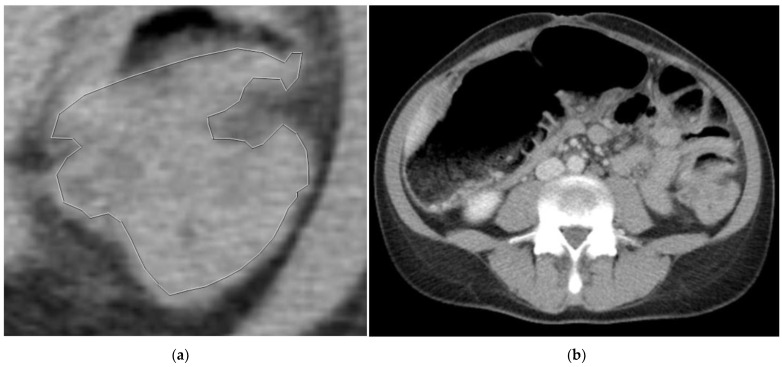
(**a**) Manually segmented non-mutated *KRAS* tumor. (**b**) Abdominal contrast-enhancement CT of the same patient.

**Figure 4 biomedicines-11-02144-f004:**
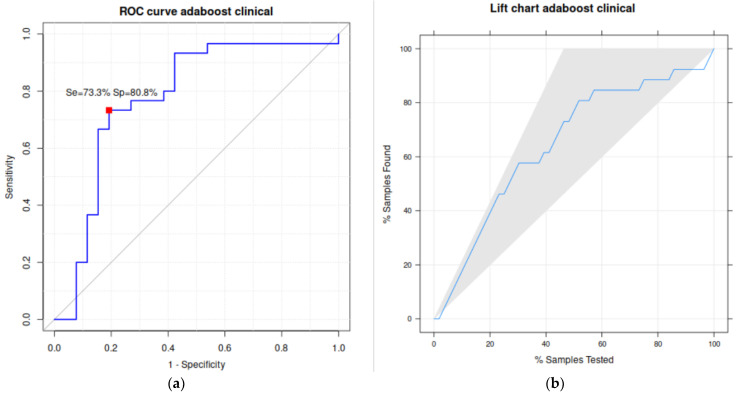
Graphical representation for the best prediction of *KRAS* mutation: classifier AdaBoost with the *clinical* vector as input. (**a**) ROC curve, with the working point in red, sensitivity, and specificity values. (**b**) Lift chart.

**Table 1 biomedicines-11-02144-t001:** List of classifiers with their implementation language, function, and module/package used and values used for hyper-parameter tuning (the notation 1:2:5 means values from 1 to 5 with step 2).

Family	Classifier	Language	Function (Module): Hyperparameter Tuning (If Any)
Discriminant Analysis	lda: linear discriminant analysis	Octave	Function train_sc with option LD2, package NaN
		Matlab	Function fitcdiscr
		Python	Function LinearDiscriminantAnalysis, module sklearn.discriminant_analysis
		R	Function lda, package MASS
	dlda: diagonal LDA	Matlab	Function fitcdiscr, option DiscrimType = diaglinear
	qda: quadratic discriminant analysis	Matlab	Function fitcdiscr, option DiscrimType = pseudoquadratic
	kfd: kernel Fisher discriminant	Python	Function Kfda, package kfda ^1^
Neural Network	mlp: multilayer perceptron	Matlab	Function fitcnet, *n_h_* = 10, *h*_1_ = max(1,⌊*N/(I + C)*⌋), *h*_0_ = max(1,⌊*h*_1_/*n_h_*⌋),Δ = max(1,(*h*_1_-*h*_0_)/*n_h_*), *h* (number of hidden neurons) = *h*_0_:Δ:*h*_1_ *N* = no. training patterns, *I* = no. features
		Python	Function MLPClassifier, module sklearn.neural_network, same *h*
	nnet: multilayer perceptron	R	Function nnet, package nnet, same *h*, weight decay = {0, 0.0001, 0.001, 0.01, 0.1}
	neuralnet: multilayer perceptron	R	Function neuralnet, package neuralnet, same *h*
	elm: extreme learning machine	Octave	Ad hoc implementation, *h* (hidden neurons)*:* 20 values in 1.⌊*N*/(*I* + *C*)⌋
Support Vector Machine	svm	Octave	LibSVM library ^2^, functions svcmtrain/svmpredict, λ (regularization) = 2^−5:2:10^, γ (RBF spread) = 2^−15:2:10^
		Python	Function SVC, module sklearn.svm, same tuning
		R	Function ksvm, module kernlab, same tuning
K-nearest neighbors	knn	Matlab	Function fitcknn, *k* (no. neighbors) = 1:2:15
		R	Function knn, package class, same *k*
Ensemble	adaboost	Matlab	Function fitcensemble, option method = AdaBoostM1 *T* (no. trees) = 10:10:50
		Python	Function AdaboostClassifier, same *T*, learning rate = 0.1:0.1:0.9
	bagging	R	Function fitcensemble, option method = Bag
	rf: random forest	Python	Function RandomForestClassifier, package sklearn.ensemble, *T* = 5:5:31, *F* (max. features) = 3:2:*I*
	gbm: gradient boosting machine	Python	Function GradientBoostingClassifier, package sklearn.ensemble, *T =* {50,100,150,200}, *D* (max. depth) = {1,3,6,9}
	avNNet: committee of neural multilayer perceptrons	R	Function avNNet, package caret, *H =* 1..9 and as nnet, decay = 0, 0.1, 0.01, 0.001, 0.0001
Regularized linear regression	lasso	Matlab	Function fitcecoc, option Learners = templateLinear with Learner = svm and Regularization = lasso or ridge, λ (regularization) = 2^−3:0.2:3^
	ridge	Matlab	
	sgd: stochastic gradient descent	Python	Function SGDClassifier, module sklearn.linear_model α (regularization) = {10*^−i^*}*_i =_* _1_^−5^, {5 · 10*^−i^*}_*i* = 1_^5^
Logisticregression	logreg	Matlab	Function mnrfit
		Python	Function LogisticRegression, module sklearn.linear_model
Decision tree	ctree: classification tree	Matlab	Function fitctree
		Python	FunctionDecisionTreeClassifier, module sklearn. tree, criterion = {Gini,entropy},splitter = {best,random}, max. features = 3, 4, *I*, *I*/4, *I*/2, $\sqrt{\left(I\right)}$, log_2_(I)
		R	Function ctree, package party, max. depth = 1.5, min. criterion = {0.01, 0.5, 0.745, 0.99}
	rpart: recursive partitioning	R	Function rpart, package rpart
Naive Bayes	nb	Matlab	Function fitcnb
		R	Function NaiveBayes, package klaR

^1^ https://github.com/concavegit/kfda, accessed on 14 June 2023; ^2^ https://www.csie.ntu.edu.tw/cjlin/libsvm, accessed on 14 June 2023.

**Table 2 biomedicines-11-02144-t002:** A list of the top 10 best combinations of feature vectors and classifiers, which achieved the highest kappa and accuracy values to predict the *KRAS* mutation.

Position	Kappa (%)	Accuracy (%)	Dataset	Classifier	Language
1	53.7	76.8	clinical	AdaBoost	Python
2	46.0	73.2	Hf_d123m4_ + clinical	mlp	Python
3	46.0	73.2	WDCF_fm_ (m = 4, f = All)	rpart	R
4	46.0	73.2	WDCF_fm_ (m = 5, f = All)	rpart	R
5	45.7	73.2	WCF_d3m5_ + clinical	ridge	Matlab
6	44.9	73.2	Hf_d3m4_ +clinical	ridge	Matlab
7	42.9	71.4	clinical	mlp	Python
8	42.6	71.4	clinical	elm	Octave
9	42.6	71.4	Hf_d2m4_ +clinical	lda	Octave
10	42.3	71.4	Hf_d2m4_ +clinical	logreg	Matlab

**Table 3 biomedicines-11-02144-t003:** Confusion matrix for the prediction of *KRAS* mutation using the clinical and imaging information (input vectors *clinical* and WDCF*_fm_*, respectively). Position 1 and 3 in Table 2.

			Computer (AdaBoost)	
Dataset			*KRAS*+	*KRAS*−
clinical vector kappa = 53.7% acc = 76.8%	Biopsy	*KRAS*+	22 (39.3%)	8 (14.3%)
		*KRAS*-	5 (8.9%)	21 (37.5%)
		**Computer (rpat)**		
WDCF*_fm_* (*f* = All) kappa = 46% acc = 73.2%	Biopsy	*KRAS*+	23 (41.1%)	7 (12.5%)
		*KRAS*-	8 (14.3%)	18 (32.1%)

**Table 4 biomedicines-11-02144-t004:** Performance metrics (in %) achieved by the best classifier (AdaBoost) and feature vector (clinical patient information).

Kappa	Accuracy	Sensitivity or Recall	Specificity	Precision or Positive Predictivity Value (PPV)	F1	AUC
53.7	76.8	73.3	80.8	81.5	77.2	77.8

**Table 5 biomedicines-11-02144-t005:** Highest kappa (in %) achieved by each family of texture descriptors, with the best classifier and its implementation language.

Texture Descriptor	Kappa(%)	Configuration	Classifier	Language
Hf*_dm_*	24.7	*d* = 123, *m* = 5	ctree	R
LPB*_R_*	35.0	*R* = 1	ctree	Python
DWT	12.0	--	dlda	Matlab
LBP*_s_*	36.9	*s* = 2	ridge	Matlab
WCF*_dm_*	28.2	*d* = 1	rpart	R
WDCF*_fm_*	46.0	*f* = All, *m* = 4	rpart	R

**Table 6 biomedicines-11-02144-t006:** Highest kappa (in %) achieved by concatenating the texture descriptor of each family with the clinical vector.

Texture Descriptor	Kappa(%)	Configuration	Classifier	Language
Hf*_dm_*	46.0	*d* = 123, *m* = 4	mlp	Python
LBB*_R_*	42.0	*R* = 3	AdaBoost	Python
DWT	34.7	--	gbm	Python
LBP*_s_*	42.0	*s* = 1	neuralnet	R
	42.0	*s* = 3	AdaBoost	Python
WCF*_dm_*	45.7	*d* = 3, *m* = 5	mlp	Python
WDCF*_fm_*	38.8	*f* = LL, *m* = 4	sgd	Python

## Data Availability

Data is unavailable due to privacy restrictions.

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
