# Peer review of "CT-Based Radiomics to Predict KRAS Mutation in CRC Patients Using a Machine Learning Algorithm: A Retrospective Study"

_biomedicines, 2023, doi:10.3390/biomedicines11082144_

Round 1

Reviewer 1 Report

The manuscript is very well written and interesting, however, here are some suggestions to improve the quality of the manuscript. 

1. In the introduction section at the end clearly state the hypotheses of your research. They can be in separate paragraph or in bullet format. Add the last paragraph describing the outline of the paper. 

2. The conclusion should be extended and contain following information: 

-> General description in one paragraph of what was done in the paper, 

-> Conclusions in the bullet form that are answers to hypotheses stated in the introduction section and summary of the discussion section 

-> The advantages and disadvantages of the proposed research methods 

-> Guideliens for the future work.

Generally, the accuracies are low. My suggestion is to utilize the oversampling methods to balance the number of samples per class. Try for example ADASYN, SMOTE, or BorderlineSMOTE method. Balance the original datasets and then apply the ML models. The imbalance between class samples has a large influence on the classification accuracy of ML models. 

The procedure should be conducted in the following way: 

1. Oversample the datasets - this will create new datasets. 

2. Train/test the ML models with these oversampled datasets, 

3. Perform the final testing on the original datasets. 

In the end, the application of these oversampling methods should improve the classification accuracy of ML models. 

Here are some papers regarding the application of oversampling methdos: 

1. https://www.mdpi.com/2076-3417/13/1/574 

2. https://www.mdpi.com/1424-8220/23/1/169

Author Response

Please see the attachment. Thak you.

Reviewer 2 Report

Major issues

1.       This manuscript presents data from the analysis of 1870 experiments (line 215).  While it explores many analysis options, issues of overfitting of data are a concern.  For each of the 34 classifiers presented in Table 1, it would be good to have the median, minimum and maximum values of kappa (arguably the most important of the modeling measures).  This could be provided in a supplementary table.

2.       Interestingly, the best kappa value was obtained using clinical data alone, with the second best model having a kappa 7.7% lower (53.7 versus 46.0).  This raises the question regarding whether the radiomics texture descriptor information is useful at all, in contrast to the claims in the conclusion.

3.       The large number of various models examined couple with the fairly small sample size should be listed as a limitation of this study.  As such, this manuscript involves model building, and lacks a strong validation step.

4.       (Table 3) The numbers in this table seem to be presented wrong.  By biopsy, there were 30 KRAS+ and 26 KRAS- patients.  However, the counts sum to those numbers for the rows, not the columns.  Please correct or explain.

5.       From Figure 2, one can see that the reported sensitivity and specificity values are incorrect.  For the point at which the sensitivity is first above 70%, the specificity is clearly less than 80%.  Please correct.

6.       (lines 253-4) From Table 2, only 2 of the top 6 models were listed as using “ctree” or “rpart”.  Please correct or explain

Minor issues

7.       (line 27) Change “specific” to “specificity”.

8.       (Experimental setup) Please provide the methods used to obtain the cutpoint that determines the sensitivity and specificity estimates.

9.       (line 266) Change “achived” to “achieved”.

10.   (line 305) I recommend changing “determining KRAS status” to “predicting KRAS status”.

11.   (line 318) If the KRAS mutation status is known, it is unclear why it needs to be “complemented by radiomic analysis”.

There are some minor errors, of which I identified a couple.

Round 2

Reviewer 1 Report

The manuscript is improved and can be accepted in this form. 

Author Response

Thank you for your help in improving the manuscript.

Reviewer 2 Report

Two revisions have helped to make the manuscript much stronger.  The Supplementary Table and Figure show the range of kappa values with the modeling.  As noted, ctree has the highest median kappa value.  For me, that is the method that I would consider strongest, as one cannot count on obtaining the maximum result on an analysis.  The ctree data should be checked for accuracy, as median is also the 75th percentile value, as shown in Figure 1.

1)      For the lasso(M) method, the minimum and the median values are both 0, so that is a common outcome.  It would be useful to note what percent of the runs results in a kappa of 0.  Is this model, the null model?

2)      The results in Table 3 haven’t been changed, and still seem incorrect.  On lines 89 and 90, the manuscript notes that there are “30KRAS+ and 26 KRAS- patients”, as determined by biopsy.  However, in Table 3, there are 27 KRAS+ and 29 KRAS- patients in the penultimate column, and 31 KRAS+ and 25 KRAS- patients in the final column.  Notably, the two sets of rows both have 30 KRAS+ and 26 KRAS- patients.  Thus, this reviewer still believes that the rows and columns of the data have been mixed up in Table 3.  The authors stated in their reply that “columns identify prediction performed by the classifier”; however, I see the columns labeled “Biopsy”, and the rows labeled with the classifiers (adaboost and rpart).

Author Response

Please see the attachment. Thak you very much.
